# Low-Density Unsaturated Polyester Resin with the Presence of Dual-Initiator

**DOI:** 10.3390/ma16134677

**Published:** 2023-06-28

**Authors:** Jinjian Zhu, Xiaojun Wang, Minzhuang Chen

**Affiliations:** Department of Composite Materials, College of Materials Science and Engineering, Nanjing Tech University, Nanjing 211800, China; 202061203137@njtech.edu.cn (J.Z.); 202061103111@njtech.edu.cn (M.C.)

**Keywords:** dual-initiator, low-density unsaturated polyester resin, free radical initiation, crosslinking

## Abstract

Dual-initiation is a new orientation of many studies in the curing of unsaturated polyester resin and the manufacture of low-density unsaturated polyester resin (LDUPR) composite materials. In our research, two kinds of low-temperature (40–70 °C) initiators (cyclohexanone peroxide (CYHP) and methyl ethyl ketone peroxide (MEKP)), one kind of medium-temperature (70–130 °C) initiator (tert-butyl peroxy-2-ethylhexanoate (TBPO)), and three kinds of high-temperature (≥130 °C) initiators (tert-butyl benzoate peroxide (TBPB), tert-amyl carbonate peroxide-2-ethylhexanoate (TAEC), and tert-butyl carbonate peroxide-2-ethylhexanoate (TBEC)) were applied to constitute different dual-initiators. Those dual-initiators were a low-temperature dual-initiator (CYHP/MEKP), medium-low-temperature dual-initiators (CYHP/TBPO and MEKP/TBPO), and high-temperature dual-initiators (TAEC/TBPB, TAEC/TBEC, and TBEC/TBPB). In the low-temperature and medium-low-temperature ranges, the LDUPR sample displayed the highest specific compression strength (*P_s_*) of 42.08 ± 0.26 MPa·g^−1^·cm^3^ in the presence of the MEKP/TBPO dual-initiator. In the high-temperature range, the LDUPR sample exhibited the highest specific compression strength (*P_s_*) of 43.32 ± 0.45 MPa·g^−1^·cm^3^ for the existence of the TAEC/TBPB dual-initiator. It is pointed out that the dual-initiator released more active free radicals, accelerating the initial curing time and the peak time of UPR. More active free radicals caused both high-activity (short-chain) molecules and low-activity (long-chain or intertwined) molecules in resin to cross-link, prolonging UPR’s curing process by approximately two minutes and resulting in an improvement of UPR’s cross-linking. In the presence of a dual-initiator, the integrated and planar microstructure of LDUPR samples performed uniformly distributed dimples, dispersed external forces, and enhanced samples’ specific compressive strength.

## 1. Introduction

Today, lightweight, higher-performance, and lower-cost materials have become an inevitable trend in material research alongside the increasing concern for sustainable development of resources, environmental protection, and energy saving [1,2,3]. During the process, low-density unsaturated polyester resin (LDUPR) and its composite materials are new hot points of research on thermosetting resin composites for their lightweight, high specific strength, excellent thermal and acoustic insulation, and conservation of raw materials [4,5,6].

In the preparation of unsaturated polyester resin (UPR) composite materials, the structure and performance of samples are closely associated with the curing process of UPR. Therefore, investigations into different curing processes of UPR are considered important. The cross-linking mechanism of UPR with radicals was considered to be initiated by thermal initiation or redox initiation [7,8,9]. Fu et al. found that a methyl ethyl ketone peroxide (MEKP)/cobalt naphthenate system showed characteristics of lower curing exothermic and slow crosslinking for LDUPR [10]. Zhang et al. improved the specific compressive strength (*P_s_*) of LDUPR, utilizing the synergistic action of methyl ethyl ketone peroxide (MEKP-II) and cobalt naphthenate [11]. Wang et al. pointed out that CaCO_3_ could hinder the cross-linking between UPR and styrene so that the exothermic heat of the polymerization decreased [5].

Among different curing processes of UPR at different temperatures, there were three different types of initiators, which were low-temperature (40–70 °C) initiators [4,12,13,14,15], medium-temperature (70–130 °C) initiators [16,17,18,19], and high-temperature (≥130 °C) initiators [20,21,22,23]. Regarding commercial low-temperature initiators, there are cyclohexanone peroxide (CYHP), methyl ethyl ketone peroxide (MEKP), and benzoyl peroxide (BPO), which were applied together with accelerators [12,24]. Regarding medium-temperature initiators, tert-butyl peroxy-2-ethylhexanoate (TBPO) and benzoyl peroxide (BPO) are the two typical examples [16,18]. In the range of high temperatures, tert-butyl peroxybenzoate (TBPB), peroxy ketone peroxide, and peroxy carbonate (BIC) are the three typical commercial initiators [23,25]. Currently, one single initiator is the traditional way to initiate the cross-linking of UPR [14,18,26]. However, in the presence of one single low-temperature initiator, the cross-linking of UPR is not complete due to the low treatment temperature. On the other hand, in the presence of one single high-temperature initiator, the curing process of UPR lasts longer. In contrast, the dual-initiator BPO/TBPO could shorten the molding cycle of the unsaturated polyester resin [27], and the initiator composition, which includes benzoyl peroxide, 2,4-dichlorobenzene acyl peroxide, and tert-butyl peroxybenzoate, could improve the curing degree of UPR at low temperatures [28]. Therefore, rapid and efficient cross-linking of UPR in the presence of dual initiation by different initiators is a novel project for UPR and LDUPR research, even to improve their properties.

In our research, two kinds of low-temperature initiators (cyclohexanone peroxide (CYHP) and methyl ethyl ketone peroxide (MEKP)), one kind of medium-temperature initiator (tert-butyl peroxy-2-ethylhexanoate (TBPO)), and three kinds of high-temperature initiators (tert-butyl benzoate peroxide (TBPB), tert-amyl carbonate peroxide-2-ethylhexanoate (TAEC), and tert-butyl carbonate peroxide-2-ethylhexanoate (TBEC)), were selected as essential initiators to synthesize different dual-initiators. CYHP and MEKP were synthesized as a dual-initiator at low temperatures. CYHP and TBPO, and MEKP and TBPO were synthesized as two kinds of dual-initiators at medium-low temperatures. TAEC and TBEC, TBPB and TBEC, and TAEC and TBPB were synthesized as three kinds of dual-initiators at high temperatures. Those dual-initiators were applied in the preparation of LDUPR and used to explore the dual-initiation mechanism, which was different from the previous single initiation. Firstly, the effects of every essential initiator on the gelation and cross-linking processes of UPR were investigated to produce an appropriate combination of different initiators. After that, the proper combination and proper dosage of different initiators for the orthogonal experiment of LDUPR sample preparation were identified, as was the suitable curing temperature range. Based on changes in density, compressive strength, and specific compressive strength of different LDUPR samples, which were in the presence of a dual-initiator, different curing mechanisms of UPR were explored and brought out under the treatment of low temperatures, medium-low temperatures, or high temperatures.

## 2. Materials and Methods

### 2.1. Materials

The UPR (Type of P17-902) used in the experiment was the orthophthalic polyester resin with an unsaturated polyester content of 63–66 wt%. It was produced by AOC Aliancys Resins Co., Ltd., Nanjing, China, with a viscosity of 1300–1500 mPa·s (at 25 ± 1 °C) and an acid value of 15–19 mg·KOH·g^−1^.

Two kinds of low-temperature initiators were utilized in this study. Those were methyl ethyl ketone peroxide (MEKP-Ⅱ), which was produced by Luoyang Shuangyue Curing Agent Co., Ltd., Luoyang, China, with a peroxide content of more than 33 wt%, and cyclohexanone peroxide (CYHP) with a peroxide content over than 50 wt%. CYHP was produced by AkzoNobel (Tianjin) Co., Ltd., Tianjin, China.

The medium-temperature initiator, tert-butyl peroxy-2-ethylhexanoate (TBPO), was produced by AkzoNobel (Tianjin) Co., Ltd., Tianjin, China, and the peroxide content was more than 98 wt%.

High-temperature initiators, which were butyl peroxybenzoate (TBPB), peroxy-2-ethylhexyl tert-amyl carbonate (TAEC), and tert-butyl peroxy-2-ethylhexyl carbonate (TBEC), were used in this study. Their specialties are listed below:

Tert-butyl peroxybenzoate (TBPB) was produced by AkzoNobel (Tianjin) Co., Ltd., Tianjin, China, with a peroxide content of more than 98 wt%.

Peroxy-2-ethylhexyl tert-amyl carbonate (TAEC), with a peroxide content of more than 96 wt%, was obtained from Arkema (Changshu) Chemical Co., Ltd., Changshu, China.

Tert-butyl peroxy-2-ethylhexyl carbonate (TBEC), with a peroxide content of more than 96 wt%, was also a product of Arkema (Changshu) Chemical Co., Ltd., Changshu, China.

According to the research on LDUPR material [5,29], NH_4_HCO_3_, used as a foaming agent in the study, was produced by Shanghai No.4 Reagent & H.V. Chem. Co., Ltd., Shanghai, China, with a purity greater than 99 wt%.

PMR-EZ, used as a release agent in the study, was obtained from Chem-Trend Co., Ltd., Shanghai, China.

### 2.2. Preparation of LDUPR Samples

According to GB/T 24148.2-2009(ISO 3672-2:2000) [30], LDUPR samples were prepared by a formulation of (part of weight): 100 parts resin: A part initiator x: B part initiator y: C part NH_4_HCO_3_ at a treating temperature of T °C. Values of A, B, and T were respectively obtained from the results of gel time experiments. The addition of the foaming agent NH_4_HCO_3_ was no more than 3.00 phr (per hundred of resin, phr) [10,11]. In the process of sample manufacture, the mold was coated with the release agent PMR-EZ and then heated at a rising rate of 5 °C/min from ambient temperature to a certain temperature, which is listed in the followed orthogonal experiment of preparation of LDUPR samples. After that, NH_4_HCO_3_ and different initiators were added to UPR glue in turn and were stirred for 5.0 min at a speed of 120 r/min using an agitator (JJ-1A, Changzhou Jintan Heng Feng Instrument Manufacturing Co., Ltd., Changzhou, China) until they were distributed homogeneously in resin glue. The resin glue, in the presence of the foaming agent and different initiators, was poured into the heated mold and treated at a curing temperature for 2.0 h. Finally, the samples were cooled down to room temperature and demolded. In a parallel experiment, five samples were prepared for each formulation. The controlling accuracy of the temperature is 0.2 °C.

### 2.3. Methods

#### 2.3.1. Gel Time Determination

The gel time of unsaturated polyester resin was tested using a resin reaction behavior analyzer (Gelprof 518, Wuhan Jiudi Composite Material Co., Ltd., Wuhan, China) in line with GB/T 24148.7-2014 (ISO 2535:2001, “Plastics-Unsaturated-polyester resins.”) [31]. The temperature precision of the instrument was 0.1 °C, and its operating temperature was less than 300.0 °C.

#### 2.3.2. Apparent Density Testing

The apparent density of LDUPR samples was tested in accordance with GB/T 6343-2009 (ISO 845:2006, “Cellular plastics and rubbers—Determination of apparent density.”) [32]. Each sample was cut into regular cylinders with a diameter of 60 ± 1 mm and a height of 50 ± 1 mm. Five samples were tested for each formula, and the average apparent density of the samples was calculated and obtained.

#### 2.3.3. Detection of Compressive Strength

Based on the standard of GB/T 8813-2020 (ISO 844:2014, “Rigid cellular plastics—Determination of compression properties,”) [33], cured samples were all cut into regular cylinders with a diameter of 60 ± 1 mm and a height of 50 ± 1 mm. The compressive strength of the cylindrical sample was tested by an electronic universal testing instrument (Z100, Zwick, Ruhr, Germany). The instrument’s precision was 0.5%, and the maximum pressure value was 100 kN. The ambient temperature of the experiment was 23 ± 2 °C, and the relative humidity was around 50 ± 5%. For each formula, five samples were tested, and the average compressive strength of the sample was calculated and obtained.

#### 2.3.4. Thermal Analysis

The curing process of UPR was studied by using a differential scanning calorimeter (Netzsch DSC 204, Netzsch Scientific Instruments Trading Co., Seibel, Germany). A sample weighing 5–10 mg was sealed in an aluminum crucible and in an atmosphere of nitrogen. The nitrogen flow rate was 30 mL/min. The modulation temperature of the DSC was 0.1 °C, and the sensitivity of the DSC was 0.1 μg.

The sample was analyzed by isothermal differential scanning calorimetry (DSC) at a specific temperature for 1.0 h to obtain the polymerization heat (*Q_P_*) of UPR [34]. Then the sample was cooled down to 20 °C at a cooling rate of 20 °C/min. Later, the sample temperature was increased from 20 °C to 200 °C at a heating rate of 10 °C/min, and the residual heat (*Q_R_*) of UPR was obtained [34]. The total heat (*Q_T_*) of the curing process was calculated by formula (1), and the curing degree (*α*) was calculated by formula (2) [34].
*Q*_*T*_ = *Q*_*P*_ + *Q*_*R*_(1)
*α* = *Q*_*P/*_*Q*_*T*_ × 100%(2)

#### 2.3.5. Microscopic Analysis

The bubble distribution in LDUPR was observed by using an electronic video microscope (GP-530H, Kunshan Gaopin Precision Instrument Co., Ltd., Kunshan, China). The apparent morphology of the LDUPR sample was analyzed by a scanning electron microscope (SEM, ZEISS ULTRA 55, Carl Zeiss AG, Oberkochen, Germany). In the SEM experiment, conductive tapes were pasted on the surface of the sample, and gold was sputtered for 200.0 s under a high vacuum to maintain its conductivity. The micromorphology of LDUPR’s matrix was observed under the condition of high vacuum and at the acceleration voltage of 20 kV.

## 3. Results and Discussion

### 3.1. Gel Time of Resin Glue with the Presence of One Single Initiator

Generally, a lower temperature corresponds to a longer gel time, and a higher temperature corresponds to a shorter gel time [35]; the gel time of UPR is usually controlled to between 35.0 min (the upper limit of gel time) and 23.0 min (the lower limit of gel time) in commercial applications [34,36]. According to the previous research, the foaming agent NH_4_HCO_3_ decomposed in a temperature range of 60.0 °C to 90.0 °C [29]. Therefore, the gel time of the resin glue, which was in the presence of different low-temperature (CYHP or MEKP) or medium-temperature initiators (TBPO), was detected in a temperature range of 60.0 °C to 80.0 °C. With the presence of 1.00 phr, 2.00 phr, and 3.00 phr of different high-temperature initiators (TAEC, TBEC, or TBPB), the gel time of the resin glue was also detected in a temperature range of 70.0 °C to 95.0 °C. Changes in gel time and the corresponding temperature range with different initiators are listed in Table 1. The proper dosage of the initiator corresponds to a proper gel time of between 35.0 min (the upper limit of gel time) and 23.0 min (the lower limit of gel time).

### 3.2. Gel Time of Resin Glue with the Presence of Dual-Initiator

According to Table 1, the proper temperature ranges of the gel time for resin glue with one single initiator, which was CYHP, MEKP, or TBPO, are 60.0–73.0 °C, 62.0–75.0 °C, and 65.0–78.0 °C, respectively. Therefore, according to the principle of overlap, the appropriate temperature range of the CYHP/MEKP dual-initiator was set at 62.0–73.0 °C. Similarly, CYHP/TBPO was fit for a temperature range of 65.0–73.0 °C, and MEKP/TBPO was fit for a temperature range of 65.0–75.0 °C. Since the gel time of the resin glue might be shortened with an increase in the dosage of the initiator and even lengthened with a decrease in temperature [35,37], the temperature range of gel time for the resin glue with a dual-initiator was explored and determined according to changes in the content of the dual-initiator and corresponding to the proper gel time. Table 2 shows the temperature range and the corresponding constitution of the CYHP/MEKP dual-initiator, the CYHP/TBPO dual-initiator, and the MEKP/TBPO dual-initiator, which conformed to the gel time requirements.

Based on Table 1, appropriate temperature ranges for the suitable gel time of resin glue with different high-temperature initiators can be obtained, in which TAEC corresponds to a temperature range of 75.0–90.0 °C, TBEC corresponds to a temperature range of 71.0–90.0 °C, and TBPB corresponds to a temperature range of 77.0–95.0 °C. Therefore, according to the principle of overlap, the proper temperature range of TAEC/TBPB, TAEC/TBEC, and TBEC/TBPB was set in the range of 77.0–90.0 °C, 75.0–90.0 °C, and 77.0–90.0 °C, respectively. The temperature range of the gel time for resin glue with a dual-initiator was detected and determined in the study, according to changes in the dosage of each dual-initiator, and also conforming to the proper gel time. Results of the corresponding temperature ranges and corresponding constitutions of TAEC/TBPB, TAEC/TBEC, and TBEC/TBPB, which were adapted to gel time requirements, are shown in Table 3. The decomposition reactions of TAEC, TBEC, and TBPB are illustrated in Figure 1.

Table 3 indicates that the dosage of every single initiator is 1.00 phr or 2.00 phr for the dual-initiator. However, only TBEC reaches 3.00 phr in the dual-initiator of TAEC/TBEC. It is considered that TAEC and TBEC decomposed to generate free radicals with steric hindrance (see Figure 1), which hindered the movement of free radicals and was not beneficial to initiate the cross-linking of resin. Therefore, the dosage of TBEC should be higher to match the proper gel time index from 23.0 min to 35.0 min.

### 3.3. Preparation of LDUPR Samples

In order to explore the influences of dual-initiator on the preparation of LDUPR samples, an orthogonal experiment was designed and conducted in the study. The apparent density (ρ), compressive strength (*P*), and specific compressive strength (*P_s_*) of the LDUPR sample, which were obtained from the orthogonal experiment, were detected and analyzed, and optimal parameters for the preparation of LDUPR samples were obtained under the condition of a dual-initiator.

Based on the results in Table 2 and Table 3, the constitution of the dual-initiator and its applicable treating temperature could be obtained. Furthermore, four factors, such as the curing temperature (factor A), the content of initiator x (factor B), the content of initiator y (factor C), and the content of the foaming agent NH_4_HCO_3_ (factor D), were selected as four factors in the orthogonal experiment. Usually, the content of the foaming agent NH_4_HCO_3_ was 1.00 phr to 3.00 phr [10,11]. As for the dual-initiator system of CYHP/MEKP, it could be determined that the curing temperature (factor A) range was 60.0 °C to 68.0 °C in intervals of 2.0 °C, the content of CYHP (factor B) or MEKP (factor C) was both 1.00 phr to 2.00 phr in intervals of 0.25 phr, and the addition of NH_4_HCO_3_ (factor D) was 1.00 phr to 3.00 phr in intervals of 0.50 phr. An orthogonal experiment of *L*_25_(5^4^), where *L* was the code name, number 4 represented the number of factors, number 5 was the levels of each factor, and number 25 represented the serial number of samples, was designed and performed. Results of the orthogonal experiment of CYHP/MEKP are illustrated in Table 4, and the optimal parameters for the preparation of LDUPR samples can be deduced. 

The results of the *L*_25_(5^4^) orthogonal experiment were analyzed by range analysis. Under a level of a certain factor, *k*_1_, *k*_2_, and *k*_3_ are average values of *ρ*, *P*, and *P_s_*, respectively. *R*_1_, *R*_2_, and *R*_3_ are ranges corresponding to *k*_1_, *k*_2_, and *k*_3_, respectively. Calculated values of *k* and *R* are shown in Table 5.

Regarding types of lightweight and high-strength material, specific compressive strength (P_s_) is the comprehensive mechanical index for LDUPR samples. Table 5 indicates that the effects of four factors on R_3_ (the range between *k*_3max_ and *k*_3min_ for *P_s_*) are the content of NH_4_HCO_3_ (factor D), the content of CYHP (factor B), the curing temperature (factor A), and the content of MEKP (factor C) in sequence. Among them, factor D corresponds to a value of 8.74 MPa·g^−1^·cm^3^, indicating the most critical factor for the specific compressive strength of LDUPR samples. Table 5 also illustrates that the specific compressive strength (*P_s_*) of sample 14 reaches the highest value of 40.06 ± 0.45 MPa·g^−1^·cm^3^ (which corresponds to an apparent density of 0.41 ± 0.02 g·cm^−3^ and a compressive strength of 16.53 ± 0.22 MPa) under the conditions of 1.75 phr CYHP, 1.00 phr MEKP, 2.00 phr NH_4_HCO_3_, and a curing temperature of 64.0 °C.

Similar to the above orthogonal experiment, *L*_25_(5^4^) orthogonal experiments with different kinds of dual-initiators and different levels of those four factors were processed for the preparation of LDUPR samples. Those included the CYHP/TBPO dual-initiator system, MEKP/TBPO dual-initiator system, TAEC/TBPB dual-initiator system, TAEC/TBEC dual-initiator system, and TBEC/TBPB dual-initiator system, respectively. The corresponding optimum preparation parameters and mechanical properties obtained from those orthogonal experiments are listed in Table 6.

Comparing the mechanical properties of samples in the presence of CYHP/MEKP, CYHP/TBPO, or MEKP/TBPO in Table 6, it is obvious that the LDUPR sample prepared in the presence of the MEKP/TBPO dual-initiator system exhibits the highest specific compressive strength (*P_s_*) of 42.08 ± 0.26 MPa·g^−1^·cm^3^, corresponding to an apparent density of 0.40 ± 0.01 g·cm^−3^ and a compressive strength of 16.90 ± 0.26 MPa. In the same way, among TAEC/TBPB, TAEC/TBEC, and TBEC/TBPB dual-initiator systems, the LDUPR sample in the presence of the TAEC/TBPB dual-initiator presents the highest specific compressive strength of 43.32 ± 0.45 MPa·g^−1^·cm^3^, corresponding to a density of 0.36 ± 0.01 g·cm^−3^ and a compressive strength of 15.60 ± 0.27 MPa. It was revealed that the compressive strength (*P*) or the specific compressive strength (*P_s_*) of the LDUPR sample prepared with the MEKP/TBPO dual-initiator was 27.8% or 24.1% higher than that of the LDUPR sample with one single initiator under low or medium temperatures [4]. Under the conditions of high temperatures, the compressive strength (*P*) or specific compressive strength (*P_s_*) of the LDUPR sample with TAEC/TBPB was 13.6% or 21.3% higher than that of the LDUPR sample with one single high-temperature initiator [19].

### 3.4. Influences of Dual-Initiator on Curing Degree of LDUPR Samples

In order to explore differences in the curing characteristics of UPR between one single initiator and a dual-initiator, the curing degree of UPR in preparations of different LDUPR samples was characterized by DSC analysis. The resin glue samples for the DSC test (corresponding to the optimum parameters in Table 6) are listed in Table 7. Isothermal DSC curves of UPR glue in the presence of MEKP, TBPO, MEKP/TBPO, TAEC, TBPB, and TAEC/TBPB are shown in Figure 2. Exothermic peak parameters obtained from isothermal DSC curves in Figure 1 are listed in Table 8. The curing degrees of those samples can be calculated and are shown in Table 8. The decomposition reactions of MEKP and TBPO are presented in Figure 3.

As shown in Table 8, the initial curing time and peak time occur at 18.9 min and 28.7 min, respectively, for sample A curing curve. However, the initial curing time and the peak time shifted to 13.5 min and 24.6 min, respectively, for sample C, which was in the presence of the MEKP/TBPO dual-initiator. Similarly, the initial curing time and the peak time of sample F are earlier than those of the sample D or sample E. It was revealed that the rate of free radical generation for a dual-initiator is higher than that for one single initiator in the unit mass of resin and unit time. Therefore, the dual-initiator can initiate the polymerization earlier.

According to Table 7 and Figure 3, in the initiation process, MEKP generates 4.00 phr radical *d*_1_, TBPO generates 1.25 phr radical *e*_1_ and 1.25 phr radical *e*_2_, and MEKP/TBPO is the sum of them. Similarly, based on Table 7 and Figure 1, TAEC generates 1.50 phr radical *a*_1_ and 1.50 phr radical *a*_2_, and TBPB generates 1.00 phr radical *c*_1_ and 1.00 phr radical *c*_2_. It is obvious that the number of free radicals released by TAEC/TBPB is more than those released by TAEC or TBPB in per unit mass resin. Table 8 indicates that the curing process of sample C is 25.6 min, which is 1.5 min longer than that of integrating with sample A and sample B (the mean value of them is 24.1 min). It is attributed to the fact that short-chain molecules in high reactivity in resin glue were initiated rapidly and cross-linked, and then long-chain molecules and intertwined polyester, which are in low reactivity, could be initiated continually by more remaining free radicals that were released by the dual-initiator. The steric hindrance of those low-reactivity molecules was non-negligible, and it prolonged the subsequent curing process of resin [38,39]. Similarly, the curing process of sample F (which is 24.7 min, as shown in Table 8) was 2.5 min longer than that of integrating with sample D and sample E (the mean value of them is 22.2 min).

The polymerization heat (*Q_P_*) of sample A in the presence of MEKP and sample B in the presence of TBPO was 117.6 J/g and 132.2 J/g, respectively, which are both lower than that of sample C (165.8 J/g) prepared by the MEKP/TBPO dual-initiator. Similarly, the *Q_P_* values of sample D (150.1 J/g) and sample E (139.8 J/g) are also both lower than that of sample F (169.5 J/g). Furthermore, in the presence of a dual-initiator, the residual heat (*Q_R_*) of UPR is lower than that of the sample initiated by one single initiator. Consequently, the curing degree of the LDUPR sample in the presence of a dual-initiator is higher than that of the LDUPR sample with one single initiator.

It is confirmed that the earlier initiation of UPR’s cross-linking curing reaction and the improvement in UPR’s curing degree are unique for the dual-initiator in light of the above analysis, which is rather different from the curing characteristics of one single initiator. Due to the decomposition of the dual-initiator producing more free radicals, both short-chain molecules and long-chain molecules in the resin could participate in polymerization to a greater content. The curing degree of UPR increases, forming a more compact spatial structure cross-linking (see Figure 4). In contrast, one single initiator could make UPR partially involved in cross-linking, resulting in a low degree of curing of resin (see Figure 5).

### 3.5. Analysis of the Microstructure of LDUPR Prepared by Dual-Initiation System

In order to study the bubble distribution and matrix morphology of LDUPR samples prepared by one single initiator or a dual-initiator, bubbles in the LDUPR samples were observed by a high-definition microscope at a magnification of 45×, and the matrix morphology of LDUPR samples was observed by a scanning electron microscope (SEM) at magnifications of 1000× and 3000×, respectively.

According to Table 7, LDUPR samples in the presence of MEKP (sample A), TBPO (sample B), and MEKP/TBPO (sample C) were all prepared at a temperature of 66.0 °C. The micrographs of LDUPR samples of sample A, sample B, and sample C are presented in Figure 6. The corresponding apparent density (*ρ*), compression strength (*P*), and specific compression strength (*P_s_*) of those LDUPR samples are shown in Table 9.

Figure 6a,d show that under the conditions of one single initiator, the bubble distribution in sample A and sample B is inhomogeneous, accompanying various-sized bubbles and linked bubbles. Regarding sample C in the presence of a dual-initiator, the uniformity of bubble size and its distribution are significantly better than those of sample A and sample B (see Figure 6g).

SEM images of sample A (see Figure 6b and its high resolution image (Figure 6c)) and sample B (see Figure 6e and its high resolution image (Figure 6f)) indicate that micro-pores and micro-cracks exist in the resin matrix and there are no plate microstructures existing. The staggered distribution of dimples and micro-cracks is obvious in the matrix of sample A or sample B. Regarding the sample C in the presence of the MEKP/TBPO dual-initiator, the micro-structure presents the characteristics of integration and planarization. Dimples are uniformly distributed in the resin matrix of sample C (see Figure 6h and its high resolution image (Figure 6i)). The microstructure schematic diagrams of two LDUPR matrices, which are illustrated in Figure 7a,b, and the external force transmitting in different LDUPR matrices are illustrated in Figure 7(a1,b1). It is concluded that defects in the micro-structure were adverse in the dispersion of an external force for the sample in the presence of one single initiator. Micro-cracks and micro-pores were destroyed by an external force and then crumbled, resulting in the resin matrix easily rupturing (see Figure 7(a1)). Dimples are favorable for dispersing an external force and enhancing the sample’s mechanical properties (see Figure 7(b1)). As a result, the compressive strength of sample C is 16.90 ± 0.26 MPa, and its specific compressive strength is 42.08 ± 0.26 MPa·g^−1^·cm^3^, which are both much higher than those of sample A and sample B.

Similarly, sample D, sample E, and sample F were all prepared at 78.0 °C according to the conditions listed in Table 7. The micrographs of those LDUPR samples in the presence of TAEC, TBPB, or TAEC/TBPB are presented in Figure 8. Their corresponding apparent density (*ρ*), compressive strength (*P*), and specific compressive strength (*P_s_*) of LDUPR samples are shown in Table 10.

Figure 8a,d indicate that linked bubbles are both in sample D and in sample E. Bubbles are different sizes and inhomogeneous. The surface of the resin matrix is not only rough but also has apparent micro-cracks, dimples, and some micro-pores (see Figure 8b and its high resolution image (Figure 8c), as well as Figure 8e and its high resolution image (Figure 8f)). The microstructural characteristics of sample D or sample E are similar to that in Figure 7a. Micro-cracks and micro-pores might be destroyed by external forces, and then cracks in the resin matrix appeared, resulting in a decrease in the mechanical properties of LDUPR samples (see Figure 7(a1)). As for sample F, bubbles are uniform in size and distributed homogenously (see Figure 8g). There are few micro-cracks and micro-pores in the resin matrix (see Figure 8h and its high resolution image (Figure 8i)). Furthermore, dimples are uniformly distributed in the resin matrix, which is beneficial for improving the mechanical properties of the LDUPR sample (see Figure 7(b1)). The features of sample F show a similar microstructure to that in Figure 7b. The compressive strength of sample F is 15.60 ± 0.27 MPa, and the specific compressive strength is 43.32 ± 0.45 MPa·g^−1^·cm^3^, which are much higher than those of sample D and sample E.

In light of the above analysis, it can be seen that the prolongation of the curing time of the dual-initiator is beneficial for the uniform distribution of bubbles in the resin, leading to lower apparent density. However, LDUPR samples prepared by one single initiator have higher density due to the bubbles’ differences in size and the worse distribution of bubbles. LDUPR samples in the presence of a dual-initiator have a higher curing degree than those prepared by a single initiator. Therefore, the microstructural characteristics of LDUPR prepared by a dual-initiator are better than that by a single initiator, resulting in better mechanical properties of materials.

## 4. Conclusions

Six types of initiators were applied to recombine six new kinds of dual-initiators, which were the low-temperature dual-initiator CYHP/MEKP, medium-low-temperature dual-initiators CYHP/TBPO and MEKP/TBPO, and high-temperature dual-initiators TAEC/TBPB, TAEC/TBEC, and TBEC/TBPB. Low-density unsaturated polyester resin samples were prepared in the presence of different dual-initiators through orthogonal experiments. The experimental results indicated that in the presence of the MEKP/TBPO dual-initiator, the sample presented the highest specific compression strength (*P_s_*) of 42.08 ± 0.26 MPa g^−1^ cm^3^ at a medium-low temperature. In the presence of the TAEC/TBPB dual-initiator, the sample exhibited the highest specific compression strength (*P_s_*) of 43.32 ± 0.45 MPa g^−1^ cm^3^ in a high-temperature range.

Thermal analysis showed that the initial curing time and the peak time of UPR with a dual-initiator were both shifted to an earlier time compared to those with one single initiator. This was attributed to the fact that more radicals were released for the dual-initiator to accelerate the cross-linking of UPR. High-activity molecules (short-chain molecules) and low-activity molecules (including long-chain or intertwined molecules) in resin glue were all initiated by a dual-initiator to take part in cross-linking, which resulted in a longer curing process of UPR. The curing times of MEKP/TBPO and TAEC/TBPB were 25.6 min or 24.7 min, respectively, which was longer than the mean curing time of integrating MEKP and TBPO or integrating TAEC and TBPB. A longer curing process of UPR is favorable for the polymerization of polyester. As a result, the curing degree of resin in the presence of MEKP/TBPO or TAEC/TBPB was 0.78 or 0.76, respectively, which was higher than that in the presence of a single initiator.

The microstructural analysis showed that bubbles in the LDUPR sample in the presence of a dual-initiator were distributed homogeneously and of a similar size. The resin matrix presented specialties of integration and planarization accompanying well-distributed dimples, which were assigned to the existence of the dual-initiator. The unique microstructure was beneficial for the elimination of external forces. Differently from previous LDUPR samples with micro-cracks and micro-pores, which were initiated by a single initiator, the physical properties of those samples initiated by a dual-initiator improved drastically.

## Figures and Tables

**Figure 1 materials-16-04677-f001:**
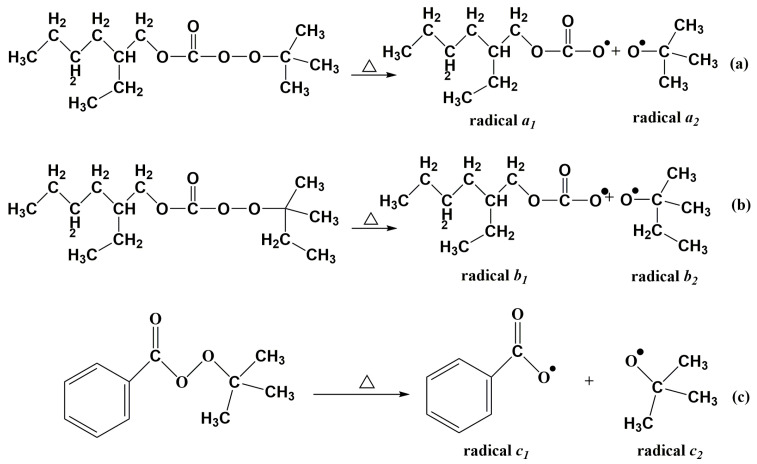
Decomposition reactions of initiators, (**a**) decomposition of TAEC, (**b**) decomposition of TBEC, (**c**) decomposition of TBPB, (△ representing heating; ● representing single electron radical).

**Figure 2 materials-16-04677-f002:**
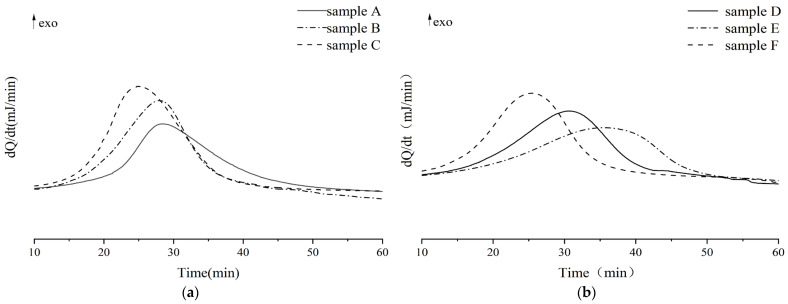
Isothermal DSC curve of (**a**) wherein sample A is cured by MEKP, sample B is cured by TBPO, sample C is cured by MEKP/TBPO; (**b**) wherein sample D is cured by TAEC, sample E is cured by TBPB, sample F is cured by TAEC/TBPB.

**Figure 3 materials-16-04677-f003:**
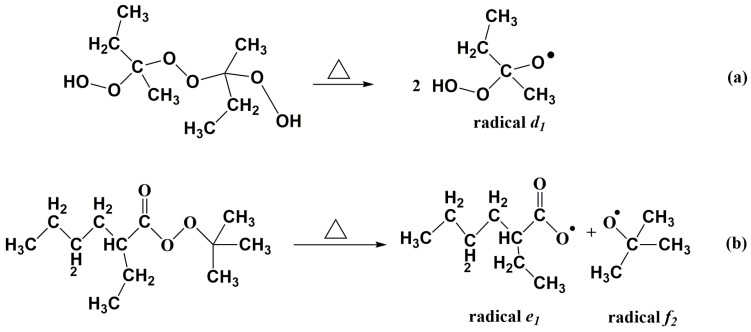
Decomposition reactions of initiators, (**a**) decomposition of MEKP, (**b**) decomposition of TBPO, (△ representing heating; ● representing single electron radical).

**Figure 4 materials-16-04677-f004:**
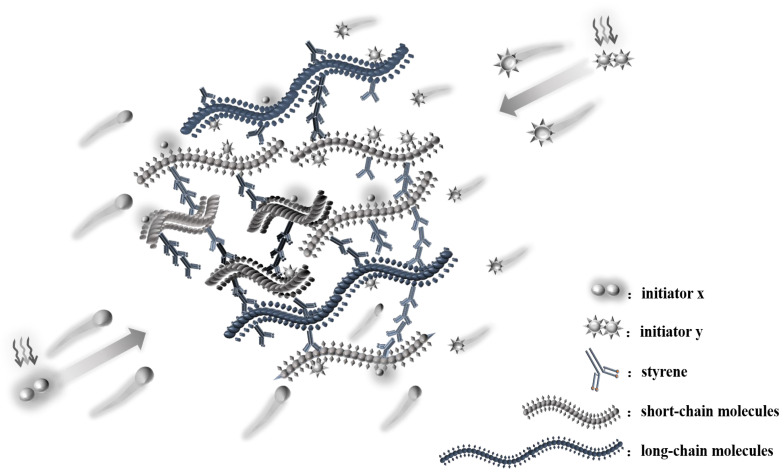
Schematic diagram of cross-linking curing of the dual-initiator.

**Figure 5 materials-16-04677-f005:**
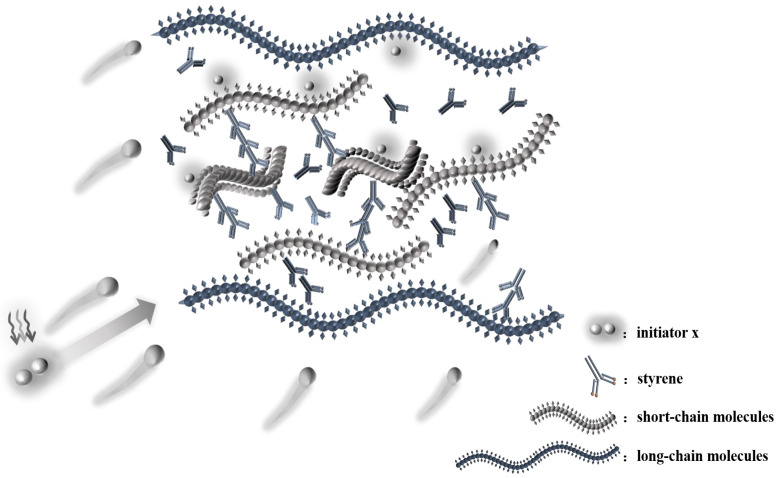
Schematic diagram of cross-linking curing of one single initiator.

**Figure 6 materials-16-04677-f006:**
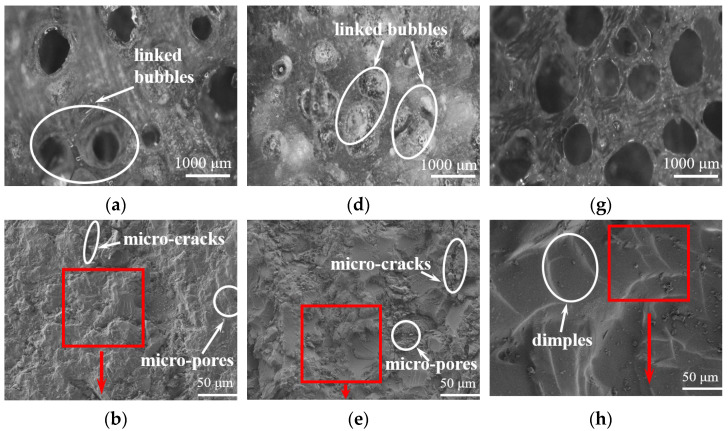
Micrographs of sample A (**a**–**c**) containing MEKP, sample B (**d**–**f**) containing TBPO, and sample C (**g**–**i**) containing MEKP/TBPO.

**Figure 7 materials-16-04677-f007:**
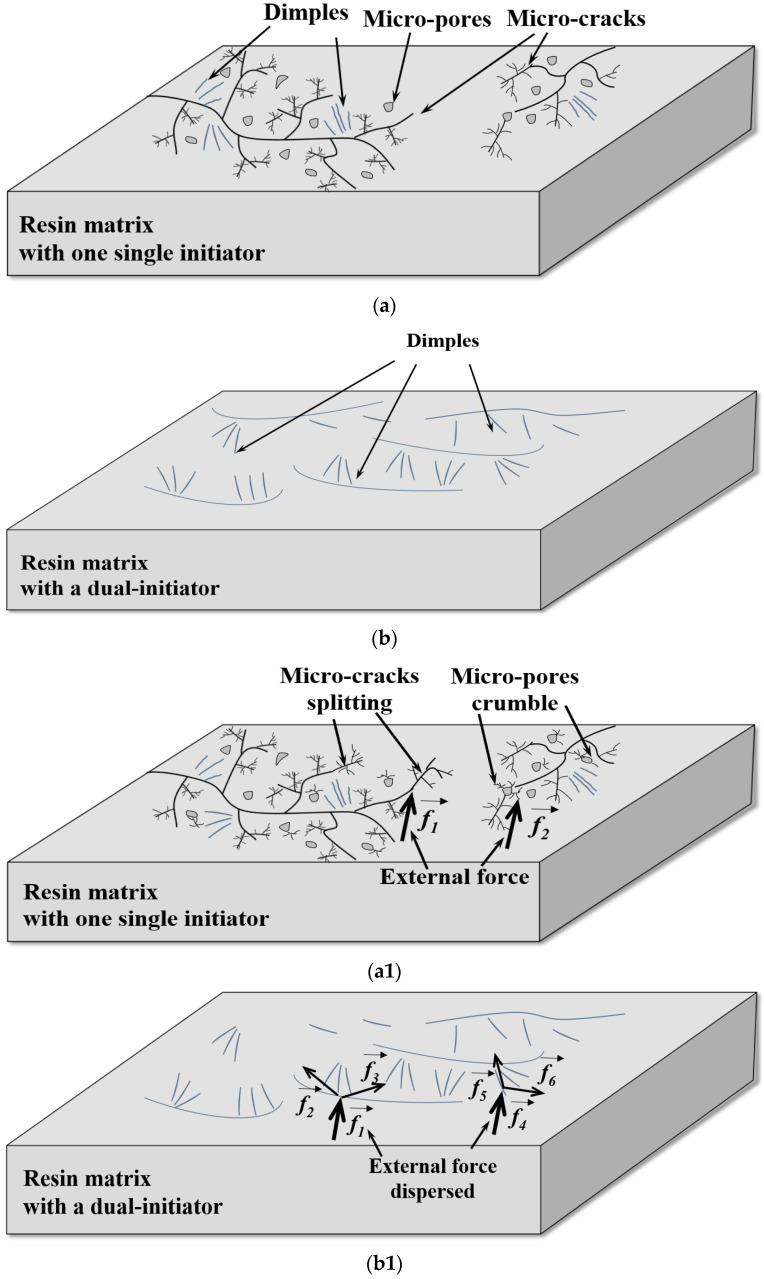
Microstructure schematic diagrams of LDUPR, (**a**) LDUPR prepared by one single initiator, (**b**) LDUPR prepared by a dual-initiator; the external force transmitting in (**a1**) LDUPR prepared by one single initiator, (**b1**) LDUPR prepared by a dual-initiator.

**Figure 8 materials-16-04677-f008:**
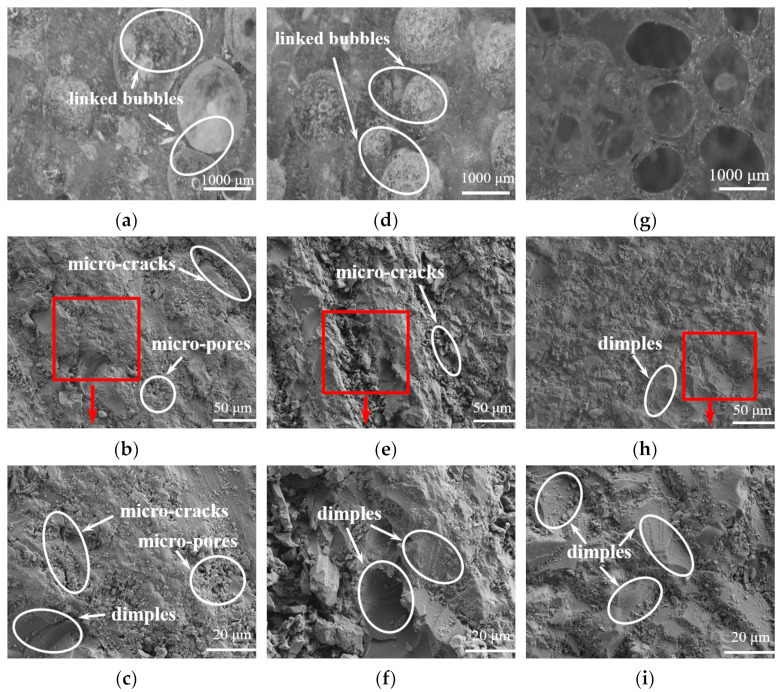
Micrographs of sample D (**a**–**c**) containing TAEC, sample E (**d**–**f**) containing TBPB, and sample F (**g**–**i**) containing TAEC/TBPB.

**Table 1 materials-16-04677-t001:** Changes in gel time and temperature range with different dosages of different initiators.

Type of Initiator	Dosage of Initiator (phr)	Temperature Range (°C)	Gel Time (min)
CYHP	1.00	65.0–73.0	34.6–23.4
	2.00	62.0–71.0	35.4–23.5
	3.00	60.0–70.0	36.0–22.4
MEKP	1.00	65.0–75.0	34.9–23.9
	2.00	63.0–73.0	34.8–23.5
	3.00	62.0–72.0	35.3–23.0
TBPO	1.00	71.0–78.0	34.8–22.5
	2.00	67.0–75.0	35.6–22.4
	3.00	65.0–73.0	36.3–23.1
TAEC	1.00	80.0–90.0	35.0–22.9
	2.00	77.0–87.0	35.4–23.1
	3.00	75.0–85.0	35.1–22.6
TBEC	1.00	73.0–90.0	35.3–23.1
	2.00	72.0–87.0	35.4–23.1
	3.00	71.0–85.0	35.2–22.8
TBPB	1.00	85.0–95.0	34.9–23.6
	2.00	82.0–93.0	34.3–23.1
	3.00	77.0–90.0	35.3–22.8

**Table 2 materials-16-04677-t002:** Temperature range and constitution of CYHP/MEKP, CYHP/TBPO, and MEKP/TBPO, which conformed to the proper gel time requirement.

Initiator Combination	Temperature Range (°C)	Component Initiator	Dosages of Component Initiators (phr)
CYHP/MEKP	60.0–68.0	CYHP	1.00, 2.00
		MEKP	1.00, 2.00
CYHP/TBPO	62.0–70.0	CYHP	1.00, 2.00
		TBPO	1.00, 2.00
MEKP/TBPO	62.0–70.0	MEKP	1.00, 2.00
		TBPO	1.00, 2.00

**Table 3 materials-16-04677-t003:** Temperature range and constitution of TAEC/TBPB, TAEC/TBEC, and TBEC/TBPB, which conformed to the gel time requirements.

Initiator Combination	Temperature Range (°C)	Component Initiator	Dosages of Component Initiators (phr)
TAEC/TBPB	72.0–80.0	TAEC	1.00, 2.00
		TBPB	1.00, 2.00
TAEC/TBEC	72.0–80.0	TAEC	1.00, 2.00
		TBEC	1.00, 2.00, 3.00
TBEC/TBPB	72.0–80.0	TBEC	1.00, 2.00
		TBPB	1.00, 2.00

**Table 4 materials-16-04677-t004:** The orthogonal experiment of CYHP/MEKP.

Sample Serial Number	Curing Temperature (°C) (A)	Content of CYHP (phr) (B)	Content of MEKP (phr) (C)	Content of NH_4_HCO_3_ (phr) (D)	*ρ* (g·cm^−3^)	*P* (MPa)	*P_s_* (MPa·g^−1^·cm^3^)
1	60.0	1.00	1.00	1.00	0.69 ± 0.02	17.66 ± 0.26	25.41 ± 0.48
2	60.0	1.25	1.25	1.50	0.58 ± 0.01	15.38 ± 0.21	26.45 ± 0.47
3	60.0	1.50	1.50	2.00	0.44 ± 0.01	13.73 ± 0.23	31.21 ± 0.45
4	60.0	1.75	1.75	2.50	0.42 ± 0.01	12.25 ± 0.14	29.17 ± 0.34
5	60.0	2.00	2.00	3.00	0.40 ± 0.02	10.87 ± 0.11	27.35 ± 0.29
6	62.0	1.00	1.25	2.00	0.41 ± 0.01	14.33 ± 0.21	35.38 ± 0.62
7	62.0	1.25	1.50	2.50	0.39 ± 0.01	12.58 ± 0.11	31.85 ± 0.45
8	62.0	1.50	1.75	3.00	0.37 ± 0.02	9.97 ± 0.13	26.95 ± 0.47
9	62.0	1.75	2.00	1.00	0.73 ± 0.03	16.32 ± 0.21	22.21 ± 0.31
10	62.0	2.00	1.00	1.50	0.63 ± 0.02	16.40 ± 0.19	25.84 ± 0.48
11	64.0	1.00	1.50	3.00	0.39 ± 0.01	8.99 ± 0.15	23.34 ± 0.26
12	64.0	1.25	1.75	1.00	0.73 ± 0.02	17.67 ± 0.31	24.37 ± 0.26
13	64.0	1.50	2.00	1.50	0.57 ± 0.01	16.68 ± 0.19	29.24 ± 0.53
14	64.0	1.75	1.00	2.00	0.41 ± 0.02	16.53 ± 0.22	40.06 ± 0.45
15	64.0	2.00	1.25	2.50	0.40 ± 0.01	10.47 ± 0.12	26.17 ± 0.47
16	66.0	1.00	1.75	1.50	0.57 ± 0.03	15.96 ± 0.29	27.79 ± 0.36
17	66.0	1.25	2.00	2.00	0.43 ± 0.01	13.85 ± 0.22	32.25 ± 0.38
18	66.0	1.50	1.00	2.50	0.38 ± 0.01	9.45 ± 0.18	24.55 ± 0.34
19	66.0	1.75	1.25	3.00	0.39 ± 0.01	9.71 ± 0.17	24.92 ± 0.34
20	66.0	2.00	1.50	1.00	0.72 ± 0.03	16.85 ± 0.31	23.24 ± 0.36
21	68.0	1.00	2.00	2.50	0.39 ± 0.01	9.36 ± 0.11	23.74 ± 0.41
22	68.0	1.25	1.00	3.00	0.34 ± 0.01	8.45 ± 0.13	24.93 ± 0.29
23	68.0	1.50	1.25	1.00	0.70 ± 0.02	18.39 ± 0.29	26.09 ± 0.32
24	68.0	1.75	1.50	1.50	0.54 ± 0.02	16.57 ± 0.18	30.73 ± 0.43
25	68.0	2.00	1.75	2.00	0.44 ± 0.02	11.39 ± 0.13	25.86 ± 0.33

**Table 5 materials-16-04677-t005:** Values of k and R of LDUPR samples in the presence of CYHP/MEKP.

Factor	Mean	Level 1	Level 2	Level 3	Level 4	Level 5	*R*_1_, *R*_2_ or *R*_3_ *(k_max_*–*k_min_)*
Curing temperature (A)	$k_{1}^{A}$ (g·cm^−3^)	0.51	0.51	0.50	0.50	0.48	0.02
	$k_{2}^{A}$ (MPa)	13.98	13.92	14.07	13.16	12.83	1.24
	$k_{3}^{A}$ (MPa·g^−1^·cm^3^)	27.92	28.45	28.69	26.55	26.27	2.42
Content of CYHP (B)	$k_{1}^{B}$ (g·cm^−3^)	0.49	0.49	0.49	0.50	0.52	0.03
	$k_{2}^{B}$ (MPa)	13.26	13.59	13.65	14.28	13.20	1.08
	$k_{3}^{B}$ (MPa·g^−1^·cm^3^)	27.13	27.97	27.61	29.47	25.69	3.77
Content of MEKP (C)	$k_{1}^{C}$ (g·cm^−3^)	0.49	0.50	0.50	0.51	0.51	0.01
	$k_{2}^{C}$ (MPa)	13.70	13.66	13.74	14.28	13.42	0.86
	$k_{3}^{C}$ (MPa·g^−1^·cm^3^)	28.20	27.80	28.08	26.83	26.97	1.38
Content of NH_4_HCO_3_ (D)	$k_{1}^{D}$ (g·cm^−3^)	0.72	0.58	0.42	0.40	0.38	0.34
	$k_{2}^{D}$ (MPa)	17.38	16.20	13.97	10.82	9.60	7.78
	$k_{3}^{D}$ (MPa·g^−1^·cm^3^)	24.26	28.02	33.00	27.10	25.50	8.74

*k*_1_: The mean of ρ calculated from five values under one level for a single factor; *R*_1_: The range between *k*_1max_ and *k*_1min_; *k*_2_: The mean of P calculated from five values under one level for a single factor; *R*_2_: The range between *k*_2max_ and *k*_2min_; *k*_3_: The mean of P_s_ calculated from five values under one level for a single factor, *R*_3_: The range between *k*_3max_ and *k*_3min_. Subscripts “A”, “B”, “C”, or “D” represent the curing temperature (factor A), the content of CYHP (factor B), the content of MEKP (factor C), and the content of NH_4_HCO_3_ (factor D), respectively.

**Table 6 materials-16-04677-t006:** Optimum preparation parameters and mechanical properties parameters of LDUPR prepared by six kinds of dual-initiator systems.

Initiation System	CYHP/MEKP	CYHP/TBPO	MEKP/TBPO	TAEC/TBPB	TAEC/TBEC	TBEC/TBPB
Temperature (°C)	64.0	66.0	66.0	78.0	76.0	76.0
Initiator x/initiator y (phr)	1.75/1.00	2.00/1.25	2.00/1.25	1.50/1.00	1.75/1.00	1.50/1.50
Content of NH_4_HCO_3_ (phr)	2.00	2.50	2.50	2.50	2.50	2.50
*ρ* (g·cm^−3^)	0.41 ± 0.02	0.41 ± 0.01	0.40 ± 0.01	0.36 ± 0.01	0.37 ± 0.02	0.37 ± 0.01
*P* (MPa)	16.53 ± 0.22	17.10 ± 0.34	16.90 ± 0.26	15.60 ± 0.27	15.34 ± 0.25	15.13 ± 0.22
*P_s_* (MPa·g^−1^·cm^3^)	40.06 ± 0.45	41.28 ± 0.26	42.08 ± 0.26	43.32 ± 0.45	41.46 ± 0.32	40.89 ± 0.29

**Table 7 materials-16-04677-t007:** Sample number and corresponding preparation parameters.

Sample Number	Curing Temperature (°C)	The Composition of the Sample
A	66.0	UPR/MEKP(2.00 phr)/NH_4_HCO_3_(2.50 phr)
B	66.0	UPR/TBPO(1.25 phr)/NH_4_HCO_3_(2.50 phr)
C	66.0	UPR/MEKP(2.00 phr)/TBPO(1.25 phr)/NH_4_HCO_3_(2.50 phr)
D	78.0	UPR/TAEC(1.50 phr)/NH_4_HCO_3_(2.50 phr)
E	78.0	UPR/TBPB(1.00 phr)/NH_4_HCO_3_(2.50 phr)
F	78.0	UPR/TAEC(1.50 phr)/TBPB(1.00 phr)/NH_4_HCO_3_(2.50 phr)

**Table 8 materials-16-04677-t008:** Exothermic peak parameters of isothermal DSC curves and curing degrees cured by MEKP, TBPO, MEKP/TBPO, TAEC, TBPB, and TAEC/TBPB.

Sample Number	Temperature (°C)	Onset (min)	Peak (min)	End (min)	Curing Time (min)	*Q_P_* (J/g)	*Q_R_* (J/g)	*Q_T_* (J/g)	*α*
A	66.0	18.9	28.7	45.1	26.2	117.6	57.6	175.2	0.67
B	66.0	16.1	27.4	38.0	21.9	132.2	52.8	185.0	0.71
C	66.0	13.5	24.6	39.1	25.6	165.8	45.9	206.7	0.78
D	78.0	18.9	30.6	41.5	22.6	150.1	63.1	213.2	0.70
E	78.0	22.5	36.2	44.2	21.7	139.8	63.9	203.7	0.69
F	78.0	12.1	25.4	36.8	24.7	169.5	54.1	223.6	0.76

**Table 9 materials-16-04677-t009:** The index of *ρ*, *P*, and *P_s_* of LDUPR samples prepared by MEKP, TBPO, and MEKP/TBPO.

Sample Number	*ρ* (g·cm^−3^)	*P* (MPa)	*P_s_* (MPa·g^−1^·cm^3^)
A	0.57 ± 0.02	14.31 ± 0.22	25.10 ± 0.49
B	0.47 ± 0.01	14.56 ± 0.20	30.99 ± 0.51
C	0.40 ± 0.01	16.90 ± 0.26	42.08 ± 0.26

**Table 10 materials-16-04677-t010:** The index of *ρ*, *P*, and *P_s_* of LDUPR samples prepared by TAEC, TBPB, and TAEC/TBPB.

Sample Number	*ρ* (g·cm^−3^)	*P* (MPa)	*P_s_* (MPa·g^−1^·cm^3^)
D	0.47 ± 0.02	13.49 ± 0.18	28.82 ± 0.53
E	0.43 ± 0.02	12.68 ± 0.14	29.36 ± 0.57
F	0.36 ± 0.01	15.60 ± 0.27	43.32 ± 0.45

## Data Availability

Not applicable.

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
