# Peer review of "Low-Density Unsaturated Polyester Resin with the Presence of Dual-Initiator"

_materials, 2023, doi:10.3390/ma16134677_

Round 1
Reviewer 1 Report
Attached

Reviewer 2 Report
The manuscript entitled "Low-Density Unsaturated Polyester Resin with The Presence of The Dual-Initiator" reports a study concerning the preparation and characterization of low-density unsaturated polyester resins in presence of different dual-initiators.
In my view, the followed experimental procedure is well designed and the manuscript is well written; therefore, I recommend the paper for publication.
My only suggestion to the Authors concerns the introduction part, which could be improved through the addition of some comments on the effect of similar dual-initiators systems already described in the literature.
Reviewer 3 Report
In this study, as stated in the title and introduction, the authors have analyzed how the presence of dual-initiators (different types and concentrations) affects the behavior of low density unsaturated polyester resin. The study is well prepared, nicely described and presented. However, I have some suggestions for the authors on how to improve the quality of their paper and also some issues need clarification.
Abstract is a little bit too long, especially the part in lines 20-27 is too detailed
Line 40, remove “as we know”-this is simply not needed
Lines 40-41, “composite materials” is repeated twice in the same sentence
Table 1, gel time, it is quite strange that the authors present the range of gel time starting from the higher value, i.e. 34.6-23.4 and not 23.4-34.6
Table 3, it should be explained more clearly in the text why solely for TAEC/TBEC – TBEC the 3phr concentration was used.
Table 4, the symbols must be explained, i.e. P, Ps
Table 4, I would strongly recommend to perform the statistical analysis of those data, i.e. does any of the factors (A, B, C) has an influence on the results?
Figure 1, wherein A cured by MEKP, it looks like this signal can be deconvoluted into two, as it is highly asymmetric. Have the Authors considered this? How can you explain the asymmetricity of this signal?
Figures 3-4, for the future studies I highly recommend some MD simulations of such systems. Those calculations can prove if the assumptions made by the Authors are correct.
Reviewer 4 Report
In the abstract you said “low-temperature” and high temperature range “ but actually you should provide these temperatures values otherwise do not have any meaning !
“Longer curing process” again what does means in terms of numerical value.
In the actual form the abstract is very vague !
Large block citation should be avoided !
“In the study, two kinds” which study ?
Also introduction contains many vague statement as the author contribution and scientific contribution is almost none
“ISO 3672-2:2000” a citation is required
“heated at T ℃.” Provide which temperature were used and also how was reached this temp !
“ISO 2535:2001” a citation is required
For any other standards the same apply as above !
“Generally, lower temperature corresponds to longer gel time” please clarify this as now it is not a research statement !
Section 3.3. belongs to part 2! Cause is a method and not a results one !
Which software were used to obtain figure 3 and 4 ?
Part of Figure 5 were repeated in Figure 7 !
A section of discussion is required
The conclusion are very qualitative some numerical details are required
Most of references are out of date please cite some novel one !
English should be considerably improved
Round 2
Reviewer 3 Report
The Authors have improved their manuscript. Therfore, this version can be accepted.
Few minor typos.
Reviewer 4 Report
It seems that the authors considered only some of my comments.
The authors do not cosnidered my advices about citation of standatds whcih I totally desagree. There are multiple version of standards and the authors should be precise !
New Figure 6 g contain the same image in new Figure 8g, so the authors should revise carfully my comments !
Also in the above mentioned figure, 6 and 8 you use differnt scale size - how can be comparble the images used in ?
I adviced about citing recent references whcih again the authors refused to do so.
n/a
